# Clinical features of behavioral symptoms in patients with semantic dementia: Does semantic dementia cause autistic traits?

**Shizuka Sakuta**[1,¤], **Mamoru Hashimoto**[2,3]*, **Manabu Ikeda**[3], **Asuka Koyama**[2], **Akihiro Takasaki**[1,4], **Maki Hotta**[1,5], **Ryuji Fukuhara**[4], **Tomohisa Ishikawa**[4], **Seiji Yuki**[4], **Yusuke Miyagawa**[4], **Yosuke Hidaka**[1], **Keiichiro Kaneda**[6], **Minoru Takebayashi**[2,7]

**1** Graduate School of Medical Sciences, Kumamoto University, Kumamoto, Japan, **2** Faculty of Life Sciences, Department of Neuropsychiatry, Kumamoto University, Kumamoto, Japan, **3** Department of Psychiatry, Osaka University Graduate School of Medicine, Suita, Osaka, Japan, **4** Department of Neuropsychiatry, Kumamoto University Hospital, Kumamoto, Japan, **5** Department of Behavioral Neurology and Neuropsychiatry, Osaka University United Graduate School of Child Development, Suita, Osaka, Japan, **6** Kumamoto Seimei Hospital, Kumamoto, Japan, **7** Division of Psychiatry and Neuroscience, Institute for Clinical Research, National Hospital Organization Kure Medical Center and Chugoku Cancer Center, Kure, Hiroshima, Japan

¤ Current address: Esaka Hospital, Suita, Osaka, Japan
* mhashimoto@psy.med.osaka-u.ac.jp

**Data Availability Statement:** All relevant data are within the manuscript.

**Funding:** This work was supported by JSPS KAKENHI Grant number JP17K10310 for MH.

## Abstract

### Objective

To investigate the behavioral characteristics of semantic dementia (SD) using an instrument originally developed for patients with autism spectrum disorder.

### Methods

The behavioral symptoms of 20 patients with SD and 20 patients with Alzheimer's disease (AD) in both the preclinical state and the dementia state were evaluated using the Pervasive Developmental Disorders Autism Society Japan Rating Scale (PARS).

### Results

The SD group showed high prevalence in four behaviors related to stereotypy and social impairment: eating very few food items, selfishness, difficulty in recognizing others' feeling and thoughts, and interpreting language literally. Scores on the PARS short version, which is sensitive for diagnosis of autism spectrum disorder, were significantly higher in the dementia state than in the preclinical state in both the SD ($11.5 \pm 6.0$ and $1.7 \pm 2.5$, respectively; $t(19) = 6.7$, $p < 0.001$) and AD ($6.9 \pm 4.6$ and $1.7 \pm 2.0$, respectively; $t(19) = 5.1$, $p < 0.001$) groups. PARS short version scores after dementia onset increased in both the SD and AD groups, although the increase was significantly larger in the SD group ($F = 5.6$, $p = 0.023$). Additionally, a significantly higher rate of patients exceeded the cutoff score for autism diagnosis in the dementia state in the SD group (75%) than in the AD group (40%; $\chi^2 = 5.0$, $p = 0.025$). PARS scores in the dementia state were significantly correlated with

https://www.jsps.go.jp/j-grantsinaid/ The funders had no role in study design, data collection and analysis, decision to publish, or preparation of the manuscript.

**Competing interests:** The authors have declared that no competing interests exist.

illness duration (r = 0.46, p = 0.04) and Mini-Mental State Examination scores (r = −0.75, p < 0.001) in the SD group only.

## Conclusions

Although SD and autism spectrum disorder are etiologically distinct diseases, patients with semantic dementia behave like those with autism spectrum disorder. Our findings suggest the symptomatic similarity of the two disorders.

## Introduction

Semantic dementia (SD) is a subtype of frontotemporal dementia (FTD) characterized by progressive aphasia caused by semantic memory loss and severe focal atrophy in the temporal lobe [1]. Although attention has been focused on verbal disabilities in SD, behavioral symptoms, which are included in the SD consensus criteria as supportive features, are also clinically important [1]. Rosen et al. compared behavioral symptoms of SD, progressive nonfluent aphasia, logopenic progressive aphasia, behavioral variant FTD (bvFTD), and Alzheimer's disease (AD) using the Neuropsychiatric Inventory (NPI) [2]. Compared with the other two progressive aphasias and AD, SD was associated with substantially more socioemotional behavioral dysfunction; specifically, more disinhibition, aberrant motor behavior, and eating disorders [2]. However, Rosen et al. found no significant differences in neuropsychiatric symptoms between SD and bvFTD, except that apathy and eating behavior were more prevalent in bvFTD than in SD patients [2]. Their findings may not fully represent the behavioral characteristics of SD because NPI is a comprehensive measure of neuropsychiatric symptoms for dementia due to Alzheimer's disease [3]. Only two studies have evaluated in detail the behavioral symptoms of SD. Snowden et al. compared clinical features of behavioral symptoms in SD and bvFTD using caregiver interviews originally designed to examine behavioral changes in FTD [4]. They found that some behavioral symptoms, including selfishness, more selective/food fads, repetitive themes, and adherence to daily routines, were more frequent in SD than in bvFTD [4]. Bozeat et al. compared behavioral symptoms among SD, bvFTD, and AD using an originally designed questionnaire to confirm the neuropsychiatric symptoms commonly reported in FTD and/or AD [5]. They showed that some behavioral symptoms, such as mental rigidity, loss of sympathy/empathy, and clockwatching, were more frequent in SD than in bvFTD and AD [5]. These studies suggest that SD may exhibit behavioral disorders distinct from those of bvFTD and AD. A recent study demonstrated that the severity of abnormal behaviors was substantially associated with caregiver burden for SD patients [6]. Therefore, it is important to clarify the clinical features of abnormal behaviors in SD.

In the present study, we investigated the behavioral characteristics of patients with SD using the Pervasive Developmental Disorders Autism Society Japan Rating Scale (PARS), which is a standardized, reliable, and valid assessment instrument for autistic traits in autism spectrum disorder (ASD). ASD is a neurodevelopmental disorder characterized by deficits in social communication accompanied by excessively repetitive behaviors, restricted interests, and insistence on sameness [7]. We used an ASD instrument for SD patients for the following reasons. First, there is a symptomatic similarity between SD and ASD. Specifically, social impairments such as selfishness and loss of empathy reported in SD correspond to a core feature of ASD, described as impairment of reciprocal social communication and social interaction [4,5,7]. Additionally, other SD behavioral symptoms, such as high frequency of complex

routines, adherence to daily routine, selective food fads, and exaggerated reaction to sensory irritation [4,5,8,9], seem to overlap with other core features and diagnostic criteria of ASD, such as repetitive behaviors, restricted interests, insistence on sameness, and hyperreactivity to sensory input [7]. Second, there is an association between neurodegeneration and myelination. Neural degeneration in FTD starts in the regions of later myelination in neural development and expands, as if to retrace the neural developmental process [10]. Thus, symptoms caused by abnormalities in neural development may present with the neural degeneration of older age. Third, temporal pole abnormalities have been reported in both SD and ASD [11–13]. Although SD and ASD are etiologically distinct diseases, assessment instruments originally developed for patients with ASD could be used to increase the understanding of behavioral symptoms in patients with SD.

We aimed to compare the clinical features of behavioral symptoms of patients with SD and those with AD using an autism rating scale. First, we evaluated behavioral symptoms of SD using the PARS to determine whether behavioral symptoms of SD comparable to those of ASD. We then investigated whether a different dementia type (i.e., AD) was also associated with autistic traits. Next, we examined whether patients with SD had exhibited autistic traits before the onset of symptoms and whether autistic trait scores increased with dementia progression.

## Materials and methods

### Participants

This study was approved by the human ethics review committee of Kumamoto University and Osaka University. Informed written consent was obtained from patients and their primary caregivers in compliance with the institution's human research standards and in accordance with the Helsinki Declaration.

Participants were selected from outpatients who attended either the dementia clinic of the Department of Neuropsychiatry at Kumamoto University Hospital or the outpatient Department of Neuropsychiatry at Osaka University Hospital between January 2018 and December 2018. Patients with SD were consecutive outpatients who fulfilled the consensus clinical diagnostic criteria developed in an international workshop for SD [1]. Patients with AD were selected from the same cohort according to the consensus criteria for the clinical diagnosis of probable AD and formed the control group [14]. Control patients were matched with SD patients for sex and illness duration. We defined illness duration as the period from the moment which the nearest caregiver was aware of the patient's cognitive or behavioral abnormality to the first assessment.

All patients received routine laboratory tests, neurological examination, and standard neuropsychological examinations, including the Mini-Mental State Examination (MMSE) [15]. Brain magnetic resonance imaging and single photon emission computed tomography of the brain were performed for all patients. All results were used for diagnoses, which were made by a team of neuropsychiatrists, neuropsychologists, and neuroradiologists. The following patients were excluded from the study: 1) those with serious psychiatric diseases such as schizophrenia, major depression, or substance abuse, and 2) those without a reliable informant.

### Assessment of neuropsychiatric symptoms

We conducted a comprehensive assessment of patients' neuropsychiatric symptoms using the Japanese version of the 12-item NPI [3]. The NPI assesses the following 12 symptoms: delusions, hallucinations, agitation/aggression, depression/dysphoria, anxiety, elation/euphoria,

apathy/indifference, disinhibition, irritability/lability, aberrant motor behavior, nighttime behavioral disturbances, and appetite/eating changes. As part of the NPI, caregivers are first asked a screening question to determine whether the symptom is present. If present, symptom frequency is rated from 1 ($<$once per week) to 4 (at least once per day), and severity from 1 (mild, present but not causing distress) to 3 (severe, very disruptive); each domain has a maximum total score (frequency × severity) of 12.

## Assessment of behavioral symptoms

To assess participant behavioral symptoms, we used the PARS, which is a semi-structured, caregiver-based interview. The PARS is a standardized tool used to assess autistic traits and obtain a diagnosis for ASD, and is widely used in Japan [16–18]. The PARS is easier to administer than current "gold standard" instruments such as the Autism Diagnostic Interview-Revised (ADI-R) [19] and the Autism Diagnostic Observation Schedule [20]. The validity of the PARS has been demonstrated by its correlation with the ADI-R [16]. There are three versions of the PARS, which target different age groups (preschoolers, primary schoolers, and adolescents/adults). Some items are common to all versions and others are specific to particular age groups. We used the adolescent/adult version in this study. The full adolescent/adult PARS comprises 33 items in 5 domains: reciprocal social interaction skills, communication skills, restricted interest, difficulty, and sensitivity [17]. The evaluator rates each item on a three-point scale (0 = none; 1 = somewhat apparent; 2 = apparent); a higher score indicates stronger ASD traits [16–18]. The PARS full version emphasizes the evaluation of difficulty and the need for administrative and medical support for ASD. In this study, we used the 33 items to qualitatively assess participants' behavioral characteristics. The short adolescent/adult version requires the selection of 12 of the 33 items that are sensitive for diagnosis of ASD and assesses three domains: reciprocal social interaction skills, communication skills, and restricted interest [18]. The PARS short version emphasizes ASD diagnosis, with ASD diagnosis cutoff scores of 7/8 [18]. We used the short version of the PARS to evaluate whether participants demonstrated autistic traits equivalent to adult ASD.

In the current study, we evaluated each participant's behavioral symptoms in both present state and preclinical state. We defined "preclinical" as the time before the onset of the symptoms. When administering the PARS, caregivers were first asked, "Does this content apply to him/her now?" for each question. To evaluate the preclinical state, caregivers were then asked, "Now, please think back to his/her state well before he/she developed some disability in language (SD patients) or memory (AD patients). Did this content apply to him/her in those days?" In principal, the PARS is administered by a physician or clinical psychologist who specializes in child and adolescent psychiatry. One psychiatrist (SS) experienced in child and adolescent psychiatry interviewed all caregivers.

## Statistical analyses

The sociodemographic and clinical characteristics of the SD and AD groups were compared using the two-tailed $t$-test for continuous data and the $\chi^2$ test and Fisher's exact test for categorical data. We compared the SD and AD groups on NPI total score and the number of patients with each neuropsychiatric symptom. Next, we compared the number of patients showing an increase in PARS score after the onset of the symptoms between the SD and AD groups using the $\chi^2$ test and Fisher's exact test for each PARS item. Additionally, we compared the PARS short version scores between the preclinical state and the present state for the SD and AD groups using the two-tailed $t$-test, paired two-tailed $t$-test, and two-way repeated measures analysis of variance. Finally, we calculated Pearson's correlation coefficients (r) for illness

**Table 1. Patient demographic and clinical characteristics.**

| | SD (n = 20) | AD (n = 20) | $t/\chi^2$ | p value |
|---|---|---|---|---|
| **Patients** | | | | |
| Age, years | 68.7 ± 5.6[a] | 70.4 ± 9.6 | 0.68 | 0.50 |
| Sex, male/female | 10/10 | 10/10 | - | - |
| Education, years | 12.9 ± 2.7 | 12.4 ± 2.2 | 0.58 | 0.56 |
| Duration of illness, years | 5.9 ± 3.0 | 5.8 ± 3.3 | 0.15 | 0.88 |
| Dominant hand, right/left/bilateral | 19/0/1 | 19/0/1 | - | - |
| MMSE score | 14.2 ± 10.1 | 16.4 ± 7.2 | 0.79 | 0.43 |
| Dominant atrophy, right/left | 6/14 | - | - | - |
| **Caregivers** | | | | |
| Relationship to patient, spouse/adult-child/other | 15/3/2 | 15/4/1 | 0.48 | 0.79 |
| Sex, male/female | 5/15 | 6/14 | 0.13 | 0.72 |

[a]Values are expressed as number or mean ± standard deviation.

duration, MMSE score, and PARS short version score to examine whether autistic traits increased with disease progression. Statistical analyses were performed using SPSS software (SPSS v.25.0; IBM Japan, Tokyo, Japan). The statistical threshold was set at p < 0.05 for each analysis.

## Results

Table 1 shows the participant characteristics. We found no significant differences between the SD and AD groups in terms of age, education, dominant hand, MMSE score, caregivers' relationship with patients, and caregiver sex.

SD, semantic dementia; AD, Alzheimer's disease; MMSE, Mini-Mental State Examination.

There was no significant difference in NPI total score between the SD (11.9 ± 13.4) and AD (11.0 ± 10.5) groups ($t$ (38) = 0.24, p = 0.81). We found no significant differences in the prevalence of individual NPI symptoms between the SD and AD groups (Table 2).

Table 3 shows the number of patients whose PARS score increased after the onset of the symptoms for each PARS item. The most frequent behavioral symptoms in the SD group were

**Table 2. Prevalence of neuropsychiatric symptoms: Comparison of SD and AD groups.**

| Each item of NPI | SD (n = 20) | AD (n = 20) | $\chi^2/t$ | p values |
|---|---|---|---|---|
| Delusions | 1 | 4 | 2.1 | 0.34 |
| Hallucinations | 0 | 2 | 2.1 | 0.49 |
| Agitation/aggression | 3 | 5 | 0.6 | 0.70 |
| Depression/dysphoria | 6 | 3 | 1.3 | 0.45 |
| Anxiety | 1 | 6 | 4.3 | 0.091 |
| Elation/euphoria | 0 | 0 | - | - |
| Apathy/indifference | 11 | 15 | 1.8 | 0.19 |
| Disinhibition | 7 | 3 | 2.1 | 0.27 |
| Irritability/lability | 6 | 4 | 0.5 | 0.72 |
| Aberrant motor behavior | 5 | 3 | 0.6 | 0.70 |
| Nighttime behavioral disturbances | 3 | 3 | 0.0 | 1.0 |
| Appetite/eating changes | 10 | 4 | 4.0 | 0.096 |

NPI, Neuropsychiatric Inventory; SD, semantic dementia; AD, Alzheimer's disease.

**Table 3. Comparison of SD and AD patients with increased PARS scores after dementia onset.**

| Each item of PARS[a] | SD (n = 20) | AD (n = 20) | $\chi^2$ | p values |
|---|---|---|---|---|
| 1. Persistently asks the same question[a] (RI) | 9 | 11 | 0.40 | 0.75 |
| 2. Becomes confused when everyday situations or routines change (RI) | 6 | 3 | 1.29 | 0.45 |
| 3. Cannot maintain personal independence due to disrupted lifestyle (R) | 3 | 9 | 4.29 | 0.082 |
| 4. Becomes unstable when recalling unpleasant memories (R) | 0 | 3 | 3.24 | 0.23 |
| 5. Extremely unbalanced diet; eats very few food items (H) | 7 | 1 | 5.62 | 0.044 |
| 6. Disturbed by particular sounds (H) | 3 | 3 | 0 | 1.0 |
| 7. Is either insensitive or oversensitive to pain, heat, etc. (H) | 13 | 8 | 2.51 | 0.21 |
| 8. Is very scared over nothing (R) | 0 | 0 | - | - |
| 9. Suddenly cries or becomes upset (R) | 0 | 1 | 1.03 | 1.0 |
| 10. Shows self-injurious action such as banging head on wall or chewing hands (R) | 2 | 0 | 2.11 | 0.49 |
| 11. Does not have appropriate friendship for his/her age (I) | 13 | 13 | 0.00 | 1.0 |
| 12. Does not care about others and behaves selfishly (I) | 10 | 3 | 5.58 | 0.04 |
| 13. Cannot respond appropriately to others (I) | 5 | 1 | 3.14 | 0.18 |
| 14. Only speaks to others when making a request (I) | 9 | 4 | 2.85 | 0.17 |
| 15. Has difficulty understanding what is said according to each situation (C) | 15 | 11 | 1.76 | 0.32 |
| 16. Tends to use difficult words without understanding their meanings (C) | 0 | 0 | - | - |
| 17. Cannot figure out who is the talker and who is the listener in conversation with many people (C) | 11 | 8 | 0.90 | 0.53 |
| 18. Cannot explain/answer with causality (C) | 14 | 15 | 0.13 | 1.0 |
| 19. Talks in monotonous and unnatural tone (C) | 2 | 0 | 2.11 | 0.48 |
| 20. Cannot recognize/understand others' feelings and thoughts (C) | 15 | 7 | 6.47 | 0.025 |
| 21. Only interprets language in a literal sense; cannot understand jokes or irony (C) | 16 | 5 | 12.13 | 0.001 |
| 22. Immerses her/himself in specific knowledge acquisition such as name of places or train stations (RI) | 3 | 0 | 3.24 | 0.07 |
| 23. Does an impression of familiar TV scene alone (RI) | 1 | 0 | 1.02 | 1.0 |
| 24. Repeats inappropriate behavior tenaciously in spite of others' disapproval (RI) | 4 | 0 | 4.44 | 0.11 |
| 25. Cannot settle down without taking first place in any situation (RI) | 1 | 0 | 1.03 | 1.0 |
| 26. Has tic disorder (R) | 1 | 2 | 0.36 | 1.0 |
| 27. Is restless and cannot adjust behavior to the occasion (R) | 8 | 3 | 3.14 | 0.16 |
| 28. Is inattentive and cannot adjust behavior to the occasion (R) | 5 | 1 | 3.14 | 0.18 |
| 29. His/her actions are often interrupted and he/she cannot transfer to the new action (R) | 5 | 5 | 0 | 1.0 |
| 30. Behaves as if s/he never feels shame (I) | 7 | 4 | 1.13 | 0.48 |
| 31. Is easily tricked (I) | 7 | 9 | 0.42 | 0.75 |
| 32. Interprets others' statements as persecutory or offensive; paranoid (R) | 2 | 4 | 0.78 | 0.66 |
| 33. Shows mood swings and recurring depression and agitation (R) | 2 | 5 | 1.56 | 0.41 |

[a]The 12 items of the PARS short version are underlined.

PARS, Pervasive Developmental Disorders Autism Society Japan Rating Scale; SD, semantic dementia; AD, Alzheimer's disease, RI; restricted interests, R; resistance, H; hypersensitivity, I; interpersonal relationship, C; communication.

"Only interprets language in a literal sense; cannot understand jokes or irony," observed in 16 patients (80%), followed by "Has difficulty understanding what is said according to each situation" (75%) and "Cannot recognize/understand others' feelings and thoughts" (75%). Compared with the AD group, the SD group showed a significantly greater score increase after the onset of the symptoms for the following four PARS items: "Extremely unbalanced diet: eats very few food items," "Does not care about others and behaves selfishly," "Cannot recognize/understand others' feelings and thoughts," and "Only interprets language in a literal sense; cannot understand jokes or irony," which could be considered SD-specific behavioral symptoms.

The PARS short version scores for the two groups in the preclinical and the present states are shown in Fig 1. The PARS short version score was significantly higher for individuals in the present state than in the preclinical state in both the SD (11.5 ± 6.0 and 1.7 ± 2.5, respectively; $t$ (19) = 6.7, p < 0.001) and AD (6.9 ± 4.6 and 1.7 ± 2.0, respectively; $t$ (19) = 5.1, p < 0.001) groups. There were no significant differences in PARS short version scores between the SD and AD groups in the preclinical state. Although the PARS short version score increased after the onset of the symptoms in both the SD and AD groups, the increase was significantly larger in the SD group than in the AD group (F = 5.6, p = 0.023). The rate of patients who exceeded the PARS cutoff score in the present state in the SD group (75%) was significantly higher than that in the AD group (40%; $\chi^2$ = 5.0, p = 0.025). Only one patient in each group (5%) exceeded the PARS cutoff score in the preclinical state.

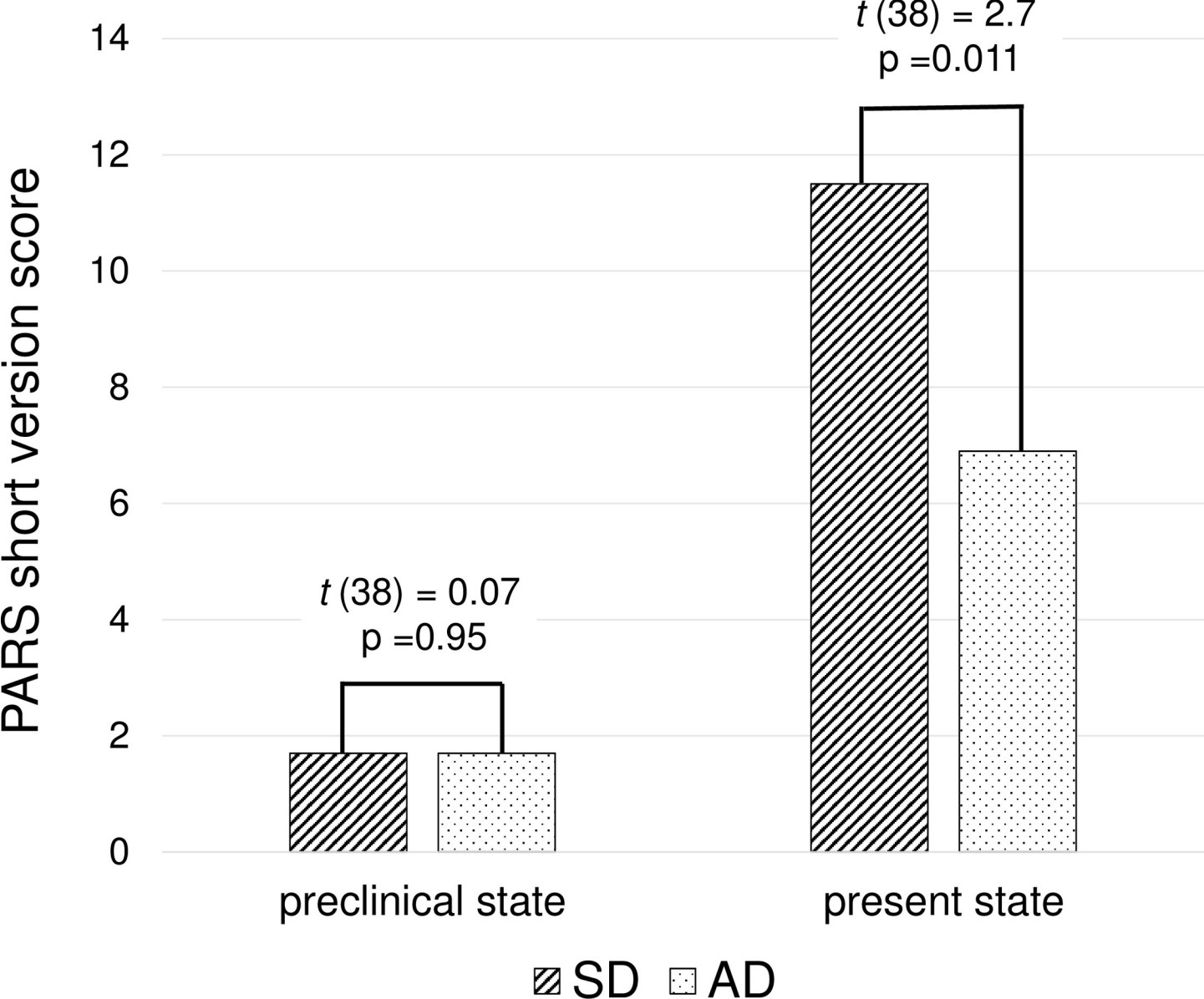

**Fig 1. PARS Scores of SD and AD patients in preclinical and present states.** Two-tailed $t$-test. PARS, Pervasive Developmental Disorders Autism Society Japan Rating Scale; SD, semantic dementia; AD, Alzheimer's disease.

A scatterplot showing the association of illness duration and MMSE score with PARS short version score in individuals in the present state is shown in Fig 2. PARS score was significantly positively correlated with illness duration (r = 0.46, p = 0.04) [a] and significantly negatively correlated with MMSE score (r = −0.75, p < 0.001) in the SD group [b]. In the AD group, we found no significant correlations between PARS score and either illness duration [c] or MMSE score [d].

## Discussion

In this study, we found no significant difference between the SD and AD group in either NPI total score or the prevalence of each neuropsychiatric symptom in NPI. This negative result

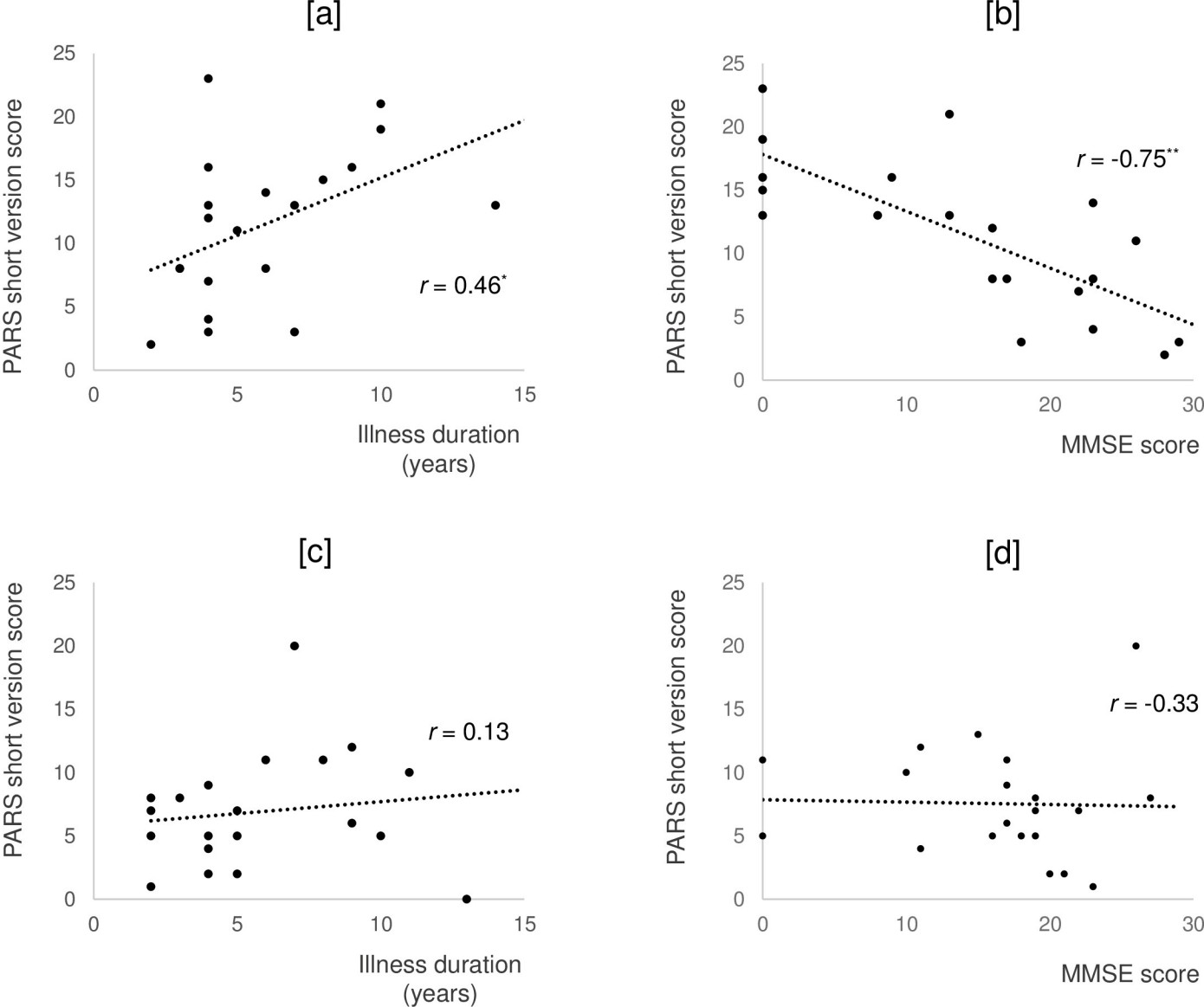

**Fig 2. Correlation of PARS Scores with illness duration and MMSE score in the present state.** [a]Correlation (Pearson's r) between PARS score and illness duration in SD group. [b] Correlation between PARS score and MMSE score in SD group. [c]Correlation between PARS score and illness duration in AD group. [d] Correlation between PARS score and MMSE score in AD group. PARS, Pervasive Developmental Disorders Autism Society Japan Rating Scale; MMSE, Mini-Mental State Examination; SD, semantic dementia; AD, Alzheimer's disease. **p < 0.01, *p < 0.05.

may partly be attributable to the small sample size. However, the increase in PARS short version score was significantly larger in the SD group than in the AD group. Furthermore, SD patients showed significant score changes for four PARS items compared with AD patients. These results demonstrate both the usefulness of the PARS and the limitation of the NPI for evaluating SD behavioral symptoms.

The primary aim of this study was to improve understanding of the behavioral disorders of SD patients by using the PARS. In this respect, it was noteworthy that three-quarters of SD patients in the present state exceeded the PARS short version cutoff score; that is, they had autistic trait levels indicative of an ASD diagnosis. Significantly more SD patients than AD patients exceeded the cutoff score, and the PARS score increase in the SD group was larger than that in the AD group. Additionally, autistic traits (as represented by the PARS score) were associated with illness duration and cognitive decline only in the SD group. These results suggest that SD onset leads to ASD-like behaviors.

According to previous studies, the prevalence of ASD in UK adults older than 16 years is 9.8 per 1000 [21], and the prevalence of ASD in US individuals aged 3–17 years is 2.24% [22]. However, the prevalence of ASD in 5-year-old children in Japan is 4.48% [23], which is comparable to our finding that only one (5%) patient exceeded the cutoff score for ASD diagnosis in a preclinical state across both the SD and AD groups. Therefore, it is likely that autistic traits in SD patients do not reflect those of individuals in a preclinical state. To our knowledge, this is the first study to mention the symptomatic similarity between SD and ASD.

We conducted a detailed evaluation of SD behavioral symptoms using the 33 PARS items. Four SD-specific behavioral symptoms were identified ("Extremely unbalanced diet; eats very few food items," "Does not care about others and behaves selfishly," "Cannot recognize/understand others' feelings and thoughts," and "Only interprets language in a literal sense; cannot understand jokes or irony"). Regarding the eating of very few food items, Snowden et al. reported a prevalence of selective/food fads of 55% in SD patients, which was more than twice that for bvFTD patients [4]. Ikeda et al. reported that the prevalence of wanting to cook or eat exactly the same foods each day was 40% in SD patients, higher than the 26% in bvFTD patients [8]. Additionally, Ikeda et al. showed the prevalence of altered food preference toward sweet foods in SD patients was equivalent to or higher than that in bvFTD patients [8]. These previous reports suggest that stereotyped behavior or changes in food preferences may cause this type of behavior in individuals with SD. Patients with ASD often eat the same food every day because of their insistence on sameness, and eat very few food items because of a hypersensitivity to food taste, smell, texture, or appearance [7]. Although the background factors for this type of eating behavior are not identical, the common feature of stereotyped behavior may generate similar behavior in both SD and ASD patients.

Social impairment (i.e., impairment of reciprocal social communication and social interaction) is a core feature of ASD and attributable to impairments in theory of mind (ToM) in ASD. In this study, SD patients presented social impairments such as selfishness and difficulty in recognizing others' feelings and thoughts. Regarding selfish behavior, Snowden et al. reported a prevalence of selfishness of 91% in SD patients, which was equivalent to or higher than that in bvFTD patients [4]. Difficulty in recognizing others' feelings and thoughts presents as a loss of empathy for others in SD. In a study examining the prevalence of changes in personality and behavior in FTD, the prevalence of lack of empathy was slightly higher in SD than in bvFTD patients [5]. Several studies show that SD patients as well as ASD patients experience problems with ToM [24,25]. Some socioemotional behavioral dysfunctions in SD that overlap with those of ASD may be related to common deficits in psychological processes such as ToM. Specific assessment instruments such as the Interpersonal Reactivity Index [26] may provide a better understanding of empathy in patients with SD in future studies.

Regarding literal interpretation of language, Rankin et al. reported that SD patients showed more impairments in simple sarcasm than bvFTD, AD, and progressive supranuclear palsy patients [27]. It is interesting that both ASD and SD patients cannot understand jokes or irony, which requires integration of a wide range of information and an understanding of the whole situation and its implications. Thus, difficulty in understanding jokes and irony is likely a result of weak central coherence, which is generally described as a limited ability in ASD patients to understand context or to see the bigger picture [28]. Like ASD patients, SD patients show under-generalization in cognitive behaviors such as semantic narrowing and an inability to see the whole picture [29]. Lambon et al. hypothesized that SD is characterized by impaired semantic generalization, in which the anterior temporal lobe plays a critical role [29]. Problems with cognitive processes such as seeing the whole picture may underlie the similarities in social behavior between SD and ASD.

Given that SD and ASD are etiologically distinct diseases, it is unsurprising that some symptomatic features were different between patients with SD and those with ASD. For example, no patients with SD showed a score change between the preclinical state and dementia state in the following PARS items: "Becomes unstable when recalling unpleasant memories", "Is very scared for no obvious reason", and "Suddenly cries or becomes upset". These PARS items evaluate the characteristic emotional instability seen in those with ASD. That said, Snowden et al. argued that a loss of basic emotion is a characteristic feature of SD [4], which indicates that people with SD and those with ASD have marked differences in their emotional expression. A recent study reported that older patients with ASD were difficult to distinguish from patients with FTD owing to symptomatic similarity [30]. The different profiles between patients with SD and ASD shown in this study may help to guide the development of strategies to improve FTD diagnosis accuracy.

The neural mechanisms underlying the behavioral symptoms of SD are still unknown. Social and behavioral disturbances observed in bvFTD are mainly explained by frontal lobe atrophy. However, numerous studies have reported that the cortical atrophy in SD is relatively localized to one side of the temporal lobes in the early stage of the disease, and that there is progressive grey and white matter involvement of the frontal and contralateral temporal lobe with disease progression [31,32]. Recently, Collins et al. examined the most prominent and consistent region of atrophy in SD patients using cortical thickness analysis [33] and found that the maximal cortical atrophy was consistently localized in the left temporal pole, whereas the progressive neurodegeneration of SD follows pathways in a large-scale neural network connected to the temporal pole [33]. No studies have directly demonstrated an association between temporal lobe atrophy and SD behavioral symptoms. However, the present finding that PARS short version scores were correlated with MMSE scores and illness duration only in the SD group suggests that neural network failure centered on the anterior temporal lobes may play a critical role in development of ASD-like behavioral disorders in SD.

Several studies have also documented structural and functional abnormalities in the temporal lobe in ASD. For example, compared with patients with typical development, ASD patients show morphological and functional changes in the temporal lobe [11–13]. Pereira et al. also reported functional changes in the default mode network between the left temporal pole and various regions in ASD [13], and the presence of thicker cortices in the right temporal pole is associated with greater communication impairment, as measured by the ADI-R [13]. The temporal pole is a key region for various social and emotional functions in typical development, such as identification and facial recognition, expression, mentalizing, and ToM [34]. Regarding the neuroanatomical implications of our study, the symptomatic similarities between SD and ASD likely reflect involvement of a common neural network centered on the anterior temporal lobes. Further study is needed to confirm the association between autistic traits and atrophy or functional decline in the temporal lobe. From the perspective of pathology, SD has been

reported to be predominantly associated with DNA-binding protein 43 (TDP-43) [35]. While the background pathology of ASD has not yet been elucidated, Cetin et al. reported that the mean serum TDP-43 levels in children with ASD were lower than those of healthy control children [36]. These findings highlight the possibility that functional abnormalities of TDP-43 may be involved in the development of behavioral symptoms that are common to both SD and ASD. However, we found no difference in autistic traits or ASD prevalence in the preclinical state between the SD and AD groups, which suggests that ASD is not a risk factor for SD. Further studies to elucidate the role of TDP-43 in both neurodegenerative and neurodevelopmental disorders would further our understanding of the neural mechanisms underlying the symptomatic similarities between SD and ASD.

There are some methodological limitations of using an assessment instrument originally developed for ASD patients to assess behavioral symptoms in dementia. First, the cognitive impairments associated with dementia may have increased PARS scores in both the SD and AD groups. For example, in the SD group, the increased scoring on the PARS item, "Has difficulty understanding what is said according to each situation," may be a result of verbal disability owing to semantic memory loss. In the AD group, more than half the patients scored higher in the present state than in the preclinical state on the PARS item, "Persistently asks the same question." However, the increased score on this item can mostly be attributed to memory impairment in the AD group. Thus, the presence of cognitive impairments may lead to an overestimation of ASD traits in patients with dementia. Second, we assessed autistic traits in the preclinical state based on the primary caregiver's memory. Prospective collection of premorbid data is extremely difficult in rare diseases such as SD. Thus, our findings in the preclinical state may have been affected by the reliability of caregiver memory. Third, the sample size was relatively small, which may have caused a type II error.

## Conclusions

The behavioral characteristics of SD were examined in detail using an instrument originally developed for ASD. We demonstrated that onset of SD produces similar behaviors to those of ASD. Our findings suggest the symptomatic similarity of SD and ASD.

## Acknowledgments

The authors thank all the patients and their families. We also thank the staff of Kumamoto University Hospital and Osaka University Hospital for their assistance. We thank Diane Williams, PhD, from Edanz Group (https://en-author-services.edanz.com/ac), for editing a draft of this manuscript.

## Author Contributions

**Conceptualization:** Shizuka Sakuta, Mamoru Hashimoto.

**Formal analysis:** Shizuka Sakuta, Mamoru Hashimoto, Asuka Koyama.

**Investigation:** Shizuka Sakuta, Manabu Ikeda, Akihiro Takasaki, Maki Hotta, Ryuji Fukuhara, Tomohisa Ishikawa, Seiji Yuki, Yusuke Miyagawa, Yosuke Hidaka, Keiichiro Kaneda.

**Supervision:** Mamoru Hashimoto, Minoru Takebayashi.

**Writing – original draft:** Shizuka Sakuta.

**Writing – review & editing:** Mamoru Hashimoto, Manabu Ikeda, Asuka Koyama, Minoru Takebayashi.

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
