## [Decision Letter · Decision Letter 0]

11 Dec 2020

PONE-D-20-33509

Clinical features of behavioral symptoms in patients with semantic dementia: Does semantic dementia cause autistic traits?

PLOS ONE

Dear Dr. Hashimoto,

Thank you for submitting your manuscript to PLOS ONE. After careful consideration, we feel that it has merit but does not fully meet PLOS ONE’s publication criteria as it currently stands. Therefore, we invite you to submit a revised version of the manuscript that addresses the points raised during the review process.

The two reviewers addressed several major and minor concerns about your manuscript. Please revise your manuscript carefully.

We look forward to receiving your revised manuscript.

Kind regards,

Kenji Hashimoto, PhD

Academic Editor

PLOS ONE

Journal Requirements:

Reviewers' comments:

Reviewer's Responses to Questions

**Comments to the Author**

1. Is the manuscript technically sound, and do the data support the conclusions?

Reviewer #1: Yes

Reviewer #2: Partly

2. Has the statistical analysis been performed appropriately and rigorously? 

Reviewer #1: Yes

Reviewer #2: No

3. Have the authors made all data underlying the findings in their manuscript fully available?

Reviewer #1: No

Reviewer #2: Yes

4. Is the manuscript presented in an intelligible fashion and written in standard English?

Reviewer #1: Yes

Reviewer #2: Yes

5. Review Comments to the Author

Reviewer #1: 1. There is a definite difference between SD and AD from a point of neuroimaging such as MRI and SPECT. Additionally, it can be methematicaly scored using VSRAD for hippocampus atrophy. The absence of these data made the manuscript confusing.

2. The rationale for the use of PARS is obscure. I do not think that the answers to the questions #12, 20 and 21 by caregivers were so important to differentiate the two dementia-related symptoms.

3. Although the authors discussed about empathy, there were no data about empathy by the IRI, QCAE, and soon. Also we did not see data about AQ.

4. The data about the correlations between PARS and MMSE is striking. But the data constitute of two groups.

5. How did authors check illness of duration?

6. There are no data about memory.

Reviewer #2: In the present study, the author compared the twenty patients with SD with the twenty patients with AD, and they found that the SD groups showed significantly higher score of autistic traits (measured by PARS) compared to the AD groups. This difference in autistic traits was enhanced when the patients in the SD group started to suffer from dementia, since the two groups were similar in terms of autistic traits prior to the onset of dementia and dementia scale (measured by NPI) at the assessment of PARS. According to the authors, this study was the first assessing SD patients using with the scale for ASD, in spite of that patients with SD have psychopathologies such as perseveration, stereotypy, impairment of ToM or other kind of characteristics, all of which were core pathology in patients with ASD.

This study was unique and the authors successfully applied the methodology of clinical research to a clinical question of certain similarities between SD and ASD in clinical practice. This topic is likely to develop as an important research area and this study may be a pioneering data in this issue. However, the following points need to be discussed and be addressed by the authors.

Point 1: The authors should consider (or hypothesize) the relationship between SD and ASD, and it would be needed to explain them in more elaborated manner particularly in the Introduction and Discussion sections. This is because one cannot come to a conclusion that both disorders have similar etiology only from similar score(s) in an assessment scale. Do the authors consider a common pathology shared by the two disorders (that is, a dimensional perspective among different diagnosis categories), or a continuous pathology between them (e.g., patient with subclinical autistic traits can develop SD)? The authors discuss more these points from the viewpoint of phenomenology and symptomatology.

Point 2: I feel that the authors emphasized too much the commonalities between SD and ASD, although SD is a type of dementia whereas ASD is a neurodevelopmental disorder. It would be better for the authors to discuss equally different profiles between them.

Point 3: Is there any reason that the authors did not describe the ethic committee of Osaka University? (although it seemed that the authors recruited patients from Osaka University)

Point 4: Is the possible score range of PARS 33 items 0-66? What is the cutoff point for this scale?

Point 5: I feel that Table 3 should generally compare patients with the mean and standard deviation of each item of PARS. Why did the authors compare this with numbers of patients exceeding some points?

Point 6: In the Discussion section, the sentence describing MRI (Page 22, Line 321) is necessary to cite the relevant reports.

Point 7: implications (Page 24, Line 345)⇒limitations?

Point 8: In the Discussion, the small number of patients in the study should be included as a limitation of study.

6. PLOS authors have the option to publish the peer review history of their article (what does this mean?). If published, this will include your full peer review and any attached files.

Reviewer #1: No

Reviewer #2: No

---

## [Author Response · Author response to Decision Letter 0]

28 Jan 2021

Editor and Reviewers’ Comments

-Editor

- There are no comments.

For Reviewer #1:

Thank you for your attentive and detailed comments. The manuscript has benefited from these insightful suggestions. 

 1. There is a definite difference between SD and AD from a point of neuroimaging such as MRI and SPECT. Additionally, it can be methematicaly scored using VSRAD for hippocampus atrophy. The absence of these data made the manuscript confusing.

Response: Thank you for this comment. While we agree that analyzing neuroimaging data, such as MRI and SPECT data, could indeed be useful for a differential diagnosis between SD and AD, the main aim of the present study was to clarify the behavioral features of SD rather than to distinguish between the two diseases. Thus, we did not statistically analyze neuroimaging data, and instead used visual inspection in light of our findings of a differential diagnosis. As you have pointed out, it is important to clarify the relationship between behavioral symptoms of SD and cerebral atrophy, so we have mentioned this as a potential topic of future research (page 24, lines 361-362).

2. The rationale for the use of PARS is obscure. I do not think that the answers to the questions #12, 20 and 21 by caregivers were so important to differentiate the two dementia-related symptoms.

Response: Thank you for highlighting this; we agree that the reason for using the PARS in this study might have been unclear. The primary aim of this study was to further our understanding of the behavioral disorders in patients with SD using the PARS. One limitation of using conventional assessment scales to assess neuropsychiatric symptoms of dementia, such as the NPI, is that they do not always adequately evaluate abnormal behavioral problems that are characteristic to SD. We reviewed the symptomatic similarities between SD and ASD reported in the existing literature, and determined that behavioral differences between SD and AD could be assessed using the PARS. Our results demonstrate that the PARS can be used to evaluate SD behavioral symptoms. In addition, the present results could help to better understand the neural mechanisms underlying behavioral symptoms of not only SD, but also ASD.

3. Although the authors discussed about empathy, there were no data about empathy by the IRI, QCAE, and soon. Also we did not see data about AQ.

Response: Thank you for this valuable comment. In this study, we used the PARS as a screening assessment scale, because our aim was to clarify the overall picture of behavioral symptoms in SD. We found that 80% of patients with SD showed a higher score for item number 20, “Cannot recognize/understand others’ feelings and thoughts”, which was indicative of a lack of empathy. However, this item only assesses one aspect of empathy; the IRI and QCAE might be needed to evaluate empathy in patients with SD in more detail in future work. Therefore, we have added the following sentence to the discussion section (page 22, line 309-311):

“Specific assessment instruments such as the Interpersonal Reactivity Index [26] may provide a better understanding of empathy in patients with SD in future studies.”

As you have pointed out, AQ is one of the major assessment instruments with which to measure autistic traits, and we did indeed use the AQ to evaluate patients’ autistic traits. However, as the AQ is self-reported assessment scale, it is considered less reliable when completed by caregivers. Moreover, it was difficult to conduct self-reported AQ for patients with dementia. In fact, 7 patients with SD had an MMSE score that was less than 10. For these reasons, we did not include the AQ results in this study.

4. The data about the correlations between PARS and MMSE is striking. But the data constitute of two groups.

Response: Thank you for this comment. As you have pointed out, we analyzed the correlation between the PARS and MMSE scores separately for the SD and AD groups. In this study, both the SD and AD groups showed a significant increase in the PARS score after the onset of dementia, although this increase was significantly larger in the SD group. This could be explained in two ways. First, the onset of any dementia could increase ASD traits, and second, SD onset could lead to ASD-like behaviors. If the increase in ASD traits results from neurodegeneration, this would be expected to result in a longer illness and a more severe dementia, and a greater increase in the PARS score. Thus, we analyzed the SD and AD groups separately to test the above two hypotheses. We found that the PARS short version scores were correlated with MMSE scores and disease duration in the SD group only. Therefore, we speculated that the ASD traits in patients with SD were a result of neurodegeneration. 

5. How did authors check illness of duration?

Response: In this study, we defined disease duration based on the interviews with caregivers. We have added the following text to the methods section to clarify this (page 8, line 123-125):

“We defined illness duration as the period from the moment which the nearest caregiver was aware of the patient’s cognitive or behavioral abnormality to the first assessment.” 

6. There are no data about memory.

Response: As you have pointed out, memory is an important background characteristic. However, patients with SD present with a marked language impairment from an early stage of the illness, and so it is difficult to evaluate memory function in patients with SD using common memory tests. Therefore, we did not perform a detailed memory test. However, given that the results of some PARS items could have been affected by patients’ memory impairments, we have mentioned this as a limitation of the present study (page 25-26, lines 378-382).

 

For Reviewer#2.

Thank you for your kind and detailed comments. The manuscript has improved with your valuable suggestions. 

Point 1: The authors should consider (or hypothesize) the relationship between SD and ASD, and it would be needed to explain them in more elaborated manner particularly in the Introduction and Discussion sections. This is because one cannot come to a conclusion that both disorders have similar etiology only from similar score(s) in an assessment scale. Do the authors consider a common pathology shared by the two disorders (that is, a dimensional perspective among different diagnosis categories), or a continuous pathology between them (e.g., patient with subclinical autistic traits can develop SD)? The authors discuss more these points from the viewpoint of phenomenology and symptomatology.

Response: Thank you for this important suggestion. We completely agree that a more detailed discussion about common pathologies shared by the two disorders was needed. In accordance with this suggestion, we have added the following sentence to the Discussion (page 24-25, lines 362-372):

“From the perspective of pathology, SD has been reported to be predominantly associated with DNA-binding protein 43 (TDP-43) [35]. While the background pathology of ASD has not yet been elucidated, Cetin et al. reported that the mean serum TDP-43 levels in children with ASD were lower than those of healthy control children [36]. These findings highlight the possibility that functional abnormalities of TDP-43 may be involved in the development of behavioral symptoms that are common to both SD and ASD. However, we found no difference in autistic traits or ASD prevalence in the preclinical state between the SD and AD groups, which suggests that ASD is not a risk factor for SD. Further studies to elucidate the role of TDP-43 in both neurodegenerative and neurodevelopmental disorders would further our understanding of the neural mechanisms underlying the symptomatic similarities between SD and ASD.”

Point 2: I feel that the authors emphasized too much the commonalities between SD and ASD, although SD is a type of dementia whereas ASD is a neurodevelopmental disorder. It would be better for the authors to discuss equally different profiles between them.

Response: Thank you for this valuable comment. We completely agree that it is also important to consider the different profiles between patients with SD and those with ASD. Therefore, we have added the following paragraph to the Discussion section (page 22-23, lines 324-335):

“Given that SD and ASD are etiologically distinct diseases, it is unsurprising that some symptomatic features were different between patients with SD and those with ASD. For example, no patients with SD showed a score change between the preclinical state and dementia state in the following PARS items: “Becomes unstable when recalling unpleasant memories”, “Is very scared for no obvious reason”, and “Suddenly cries or becomes upset”. These PARS items evaluate the characteristic emotional instability seen in those with ASD. That said, Snowden et al. argued that a loss of basic emotion is a characteristic feature of SD [4], which indicates that people with SD and those with ASD have marked differences in their emotional expression. A recent study reported that older patients with ASD were difficult to distinguish from patients with FTD owing to symptomatic similarity [30]. The different profiles between patients with SD and ASD shown in this study may help to guide the development of strategies to improve FTD diagnosis accuracy.”

Point 3: Is there any reason that the authors did not describe the ethic committee of Osaka University? (although it seemed that the authors recruited patients from Osaka University)

Response: Thank you for this important point. This was a simple mistake, and the study was approved by the human ethics review committees of both Kumamoto University and Osaka University. We have corrected this, as suggested (page 8, line 113).

Point 4: Is the possible score range of PARS 33 items 0-66? What is the cutoff point for this scale?

Response: Thank you for this valuable comment. As you have pointed out, the PARS full version, which was developed for adolescent and adults, has 33 items and a score range of 0 to 66. The cut-off scores of the PARS full version for a probable ASD diagnosis, as well as an imminent need for administrative and medical support, is 19 out of 20. Considering that the PARS full version includes items that evaluate symptoms that increase caregivers’ burden and which reflect the necessity of medical or administrative intervention, we used the PARS short version, which is more targeted to making an ASD diagnosis. 

Point 5: I feel that Table 3 should generally compare patients with the mean and standard deviation of each item of PARS. Why did the authors compare this with numbers of patients exceeding some points?

Response: Thank you for this comment. In this study, we compared the number of patients that showed a score increase between the SD and AD groups, and for two reasons. First, we asked the clinical question of how many patients with SD had ASD-like behaviors. Second, given that each item of the PARS is rated on a 3-point scale (0 = none; 1 = somewhat apparent; 2 = apparent), it was difficult to quantify the change of each item. As you have pointed out, future work could more effectively compare these changes using other assessment scales.

Point 6: In the Discussion section, the sentence describing MRI (Page 22, Line 321) is necessary to cite the relevant reports.

Response: Thank you for this thoughtful comment. In accordance with your suggestion, we have added the two following references to the Discussion section, and have added these to the reference list:

31. Whitwell JL, Josephs KA. Recent advances in the imaging of frontotemporal dementia. Curr Neurol Neurosci Rep. 2012;12: 715-723.

32. Diehl-Schmid J, Onur OA, Kuhn J, Gruppe T, Drzezga A. Imaging frontotemporal lobar degeneration. Curr Neurol Neurosci Rep. 2014;14: 489.

Point 7: implications (Page 24, Line 345)⇒limitations?

Response: In accordance with your suggestion, we have corrected this (page 25, line 373).

Point 8: In the Discussion, the small number of patients in the study should be included as a limitation of study.

Response: In accordance with your suggestion, we have added the following sentence to the limitations section (page 26, lines 386):

“Third, the sample size was relatively small, which may have caused a type II error.”

---

## [Decision Letter · Decision Letter 1]

3 Feb 2021

Clinical features of behavioral symptoms in patients with semantic dementia: Does semantic dementia cause autistic traits?

PONE-D-20-33509R1

Dear Dr. Hashimoto,

We’re pleased to inform you that your manuscript has been judged scientifically suitable for publication and will be formally accepted for publication once it meets all outstanding technical requirements.

Kind regards,

Kenji Hashimoto, PhD

Section Editor

PLOS ONE

Additional Editor Comments (optional):

Reviewers' comments:

Reviewer's Responses to Questions

**Comments to the Author**

1. If the authors have adequately addressed your comments raised in a previous round of review and you feel that this manuscript is now acceptable for publication, you may indicate that here to bypass the “Comments to the Author” section, enter your conflict of interest statement in the “Confidential to Editor” section, and submit your "Accept" recommendation.

Reviewer #2: All comments have been addressed

2. Is the manuscript technically sound, and do the data support the conclusions?

Reviewer #2: Partly

3. Has the statistical analysis been performed appropriately and rigorously? 

Reviewer #2: Yes

4. Have the authors made all data underlying the findings in their manuscript fully available?

Reviewer #2: Yes

5. Is the manuscript presented in an intelligible fashion and written in standard English?

Reviewer #2: Yes

6. Review Comments to the Author

Reviewer #2: I think that the authors addressed the points/questions I raised and that the revised version overall was improved.

7. PLOS authors have the option to publish the peer review history of their article (what does this mean?). If published, this will include your full peer review and any attached files.

Reviewer #2: **Yes: **Nobuhisa Kanahara

---

## [Editor Report · Acceptance letter]

8 Feb 2021

PONE-D-20-33509R1 

Clinical features of behavioral symptoms in patients with semantic dementia:Does semantic dementia cause autistic traits? 

Dear Dr. Hashimoto:

I'm pleased to inform you that your manuscript has been deemed suitable for publication in PLOS ONE. Congratulations! Your manuscript is now with our production department. 

Kind regards, 

on behalf of

Prof. Kenji Hashimoto 

Section Editor

PLOS ONE